# COVID-19 vaccine hesitancy and its determinants among sub-Saharan African adolescents

Dongqing Wang[1]*, Angela Chukwu[2], Mary Mwanyika-Sando[3], Sulemana Watara Abubakari[4], Nega Assefa[5], Isabel Madzorera[6], Elena C. Hemler[6], Abbas Ismail[7], Bruno Lankoande[8], Frank Mapendo[3], Ourohiré Millogo[9], Firehiwot Workneh[10], Temesgen Azemraw[10], Lawrence Gyabaa Febir[4], Christabel James[11], Amani Tinkasimile[3], Kwaku Poku Asante[4], Till Baernighausen[6,12,13], Yemane Berhane[10], Japhet Killewo[14], Ayoade M. J. Oduola[11], Ali Sie[9], Emily R. Smith[15,16], Abdramane Bassiahi Soura[8], Tajudeen Raji[17], Said Vuai[7‡], Wafaie W. Fawzi[6,18,19‡]*

1 Department of Global and Community Health, College of Health and Human Services, George Mason University, Fairfax, Virginia, United States of America, 2 Department of Statistics, University of Ibadan, Ibadan, Nigeria, 3 Africa Academy for Public Health, Dar es Salaam, Tanzania, 4 Kintampo Health Research Centre, Research and Development Division, Ghana Health Service, Kintampo North Municipality, Bono East Region, Ghana, 5 College of Health and Medical Sciences, Haramaya University, Harar, Ethiopia, 6 Department of Global Health and Population, Harvard T.H. Chan School of Public Health, Harvard University, Boston, Massachusetts, United States of America, 7 College of Natural and Mathematical Sciences, University of Dodoma, Dodoma, Tanzania, 8 Institut Supérieur des Sciences de la Population, University of Ouagadougou, Ouagadougou, Burkina Faso, 9 Nouna Health Research Center, Nouna, Burkina Faso, 10 Addis Continental Institute of Public Health, Addis Ababa, Ethiopia, 11 University of Ibadan Research Foundation, University of Ibadan, Ibadan, Nigeria, 12 Heidelberg Institute of Global Health, University of Heidelberg, Heidelberg, Germany, 13 Africa Health Research Institute, KwaZulu-Natal, South Africa, 14 Department of Epidemiology and Biostatistics, Muhimbili University of Health and Allied Sciences, Dar es Salaam, Tanzania, 15 Department of Global Health, Milken Institute School of Public Health, George Washington University, Washington, DC, United States of America, 16 Department of Exercise and Nutrition Sciences, Milken Institute School of Public Health, George Washington University, Washington, DC, United States of America, 17 Division of Public Health Institutes and Research, Africa Centres for Disease Control and Prevention, Addis Ababa, Ethiopia, 18 Department of Nutrition, Harvard T.H. Chan School of Public Health, Harvard University, Boston, Massachusetts, United States of America, 19 Department of Epidemiology, Harvard T.H. Chan School of Public Health, Harvard University, Boston, Massachusetts, United States of America

☯ These authors contributed equally to this work.
‡ SV and WWF also contributed equally to this work as last authors.
* dwang25@gmu.edu (DW); mina@hsph.harvard.edu (WWF)

**Data Availability Statement:** Individual participant data cannot be shared publicly. A data transfer agreement between Harvard T.H. Chan School of Public Health, Africa Academy for Public Health,

## Abstract

COVID-19 vaccine hesitancy among adolescents poses a challenge to the global effort to control the pandemic. This multi-country survey aimed to assess the prevalence and determinants of COVID-19 vaccine hesitancy among adolescents in sub-Saharan Africa between July and December 2021. The survey was conducted using computer-assisted telephone interviewing among adolescents in five sub-Saharan African countries, Burkina Faso, Ethiopia, Ghana, Nigeria, and Tanzania. A rural area and an urban area were included in each country (except Ghana, which only had a rural area), with approximately 300 adolescents in each area and 2662 in total. Sociodemographic characteristics and perceptions and attitudes on COVID-19 vaccines were measured. Vaccine hesitancy was defined as definitely not getting vaccinated or being undecided on whether to get vaccinated if a COVID-19

and participating institutions (including Addis Continental Institute of Public Health, Nouna Health Research Center, Muhimbili University of Health and Allied Sciences, University of Dodoma, University of Ibadan, and Heidelberg Institute of Global Health) stipulates that data will be kept confidential and will not be shared beyond the research teams without prior permission. The de-identified dataset supporting this research may be made available following a request submitted to ghp@hsph.harvard.edu and be granted after obtaining permissions from each participating institution.

**Funding:** This work was supported by institutional support from Harvard T.H. Chan School of Public Health, Boston, MA (WWF), Harvard University Center for African Studies, Boston, MA (WWF), Heidelberg Institute of Global Health, Germany (TB), and the George Washington University Milken Institute School of Public Health, Washington, DC (ERS). The funders had no role in study design, data collection and analysis, decision to publish, or preparation of the manuscript.

**Competing interests:** The authors have declared that no competing interests exist.

vaccine were available. Log-binomial models were used to calculate the adjusted prevalence ratios (aPRs) and 95% confidence intervals (CIs) for associations between potential determinants and COVID-19 vaccine hesitancy. The percentage of COVID-19 vaccine hesitancy was 14% in rural Kersa, 23% in rural Ibadan, 31% in rural Nouna, 32% in urban Ouagadougou, 37% in urban Addis Ababa, 48% in rural Kintampo, 65% in urban Lagos, 76% in urban Dar es Salaam, and 88% in rural Dodoma. Perceived low necessity, concerns about vaccine safety, and concerns about vaccine effectiveness were the leading reasons for hesitancy. Healthcare workers, parents or family members, and schoolteachers had the greatest impacts on vaccine willingness. Perceived lack of safety (aPR: 3.52; 95% CI: 3.00, 4.13) and lack of effectiveness (aPR: 3.46; 95% CI: 2.97, 4.03) were associated with greater vaccine hesitancy. The prevalence of COVID-19 vaccine hesitancy among adolescents is alarmingly high across the five sub-Saharan African countries, especially in Tanzania. COVID-19 vaccination campaigns among sub-Saharan African adolescents should address their concerns and misconceptions about vaccine safety and effectiveness.

## Introduction

With unprecedented speed, multiple vaccines against SARS-CoV-2 were developed one year after the beginning of the coronavirus disease 2019 (COVID-19) pandemic. While the COVID-19 vaccination campaigns worldwide continue with the immunization of adults, COVID-19 vaccines have been or may be made available to adolescents in many settings.

The morbidity and mortality from COVID-19 are much lower in adolescents than among adults, especially when compared to the most vulnerable group aged 65 years and older [1]. However, there is a strong rationale for providing COVID-19 vaccines to children and adolescents. Infected adolescents can still transmit the virus to other individuals [2], and some adolescents do develop severe symptoms and complications from COVID-19 [3–5]. Therefore, vaccinating the adolescent population has the dual benefits of protecting adolescents against morbidity and mortality while reducing the spread of the virus by promoting herd immunity [2, 6]. In sub-Saharan Africa, adolescents aged 10 to 19 years make up 23% of the population [7]. Therefore, getting COVID-19 vaccines into the arms of adolescents is crucial for Africa to achieve the World Health Organization's target of 70% COVID-19 vaccination coverage by mid-2022 [8]. Further, adolescents may serve as agents of advocacy that encourage their family members and friends to get vaccinated [9].

COVID-19 vaccines are safe and effective in adolescents [10]. However, studies in high-income settings show the concerning issue of COVID-19 vaccine hesitancy among adolescents. A survey conducted by the United States Centers for Disease Control and Prevention (CDC) among adolescents and their parents in April 2021 (just before the expanded availability of the Pfizer-BioNTech vaccine to adolescents) showed that only 52% of unvaccinated adolescents aged 13 to 17 years would definitely or probably receive a COVID-19 vaccine [11]. Vaccine hesitancy, defined as the reluctance to accept available vaccines [12], was considered by the World Health Organization as one of the top threats to global health even before the COVID-19 pandemic [13]. Vaccine hesitancy of adolescents poses a challenge to the global effort to control the COVID-19 pandemic.

Based on the health belief model, the likelihood of receiving COVID-19 vaccines may be affected by various factors, such as perceived threats from COVID-19 infection, evaluation of

COVID-19 vaccines, cues to action received from the media and other individuals, and modifying sociodemographic characteristics. Behaviors toward vaccines may also be driven by cultural, social, historical, and political factors, which likely vary across cultural and geographical settings [14]. As a result, COVID-19 vaccine hesitancy varies considerably across regions and countries, as shown repeatedly in previous surveys among adults [15–20]. Further, the prevalence and determinants of vaccine hesitancy in adults may not match those among adolescents [21]. Previous studies on COVID-19 vaccine hesitancy of adolescents were conducted in high-income countries [11, 21–24] and China [9], whereas studies in sub-Saharan African countries [18, 25–31] and other LMICs [29, 32–42] have primarily examined the adult population. To our knowledge, no study has specifically examined the prevalence and determinants of COVID-19 vaccine hesitancy among adolescents across diverse settings in sub-Saharan Africa.

With the COVID-19 vaccination efforts around the globe increasingly moving toward children and adolescents, a better understanding of the extent and the driving factors of vaccine hesitancy in adolescents is needed to devise vaccination campaigns that increase vaccine uptake among adolescents. In this telephone survey across five sub-Saharan African countries, we aimed to assess the prevalence of COVID-19 vaccine hesitancy in adolescents and evaluate the potential determinants of adolescent COVID-19 vaccine hesitancy.

## Methods

### Ethics statement

This study was approved by the Institutional Review Board at Harvard T.H. Chan School of Public Health (IRB20-0909) and ethical review boards in each country and area, including the Nouna Health Research Center Ethical Committee (2020-009-/MS/SG/INSP/CRSN/CIE) and National Ethics Committee in Burkina Faso (2020-7-127), the Institutional Ethical Review Board of Addis Continental Institute of Public Health in Ethiopia (ACIPH/IRB/005/2021), Kintampo Health Research Centre Institutional Ethics Committee in Ghana (KHRCIEC/2021-12), the University of Ibadan Research Ethics Committee in Nigeria (UI/SSHEC/2020/0017), and the University of Dodoma (No. MA.84/261/02/134), Muhimbili University of Health and Allied Sciences (No. DA 282/298/06/C/767), and National Institute for Medical Research in Tanzania (NIMR/HQ/R.8a/Vol. IX/3775). Verbal parental consent and adolescent assent were obtained for adolescents younger than 18 years of age; oral informed consent was obtained from adolescents aged 18 years and older.

### Study design and study population

This cross-country study was based on a survey that used a novel mobile platform and computer-assisted telephone interviewing to collect data from sub-Saharan African adolescents, adults, and healthcare providers. In the previous round of the survey (Round 1), six areas from three countries were included, namely Nouna and Ouagadougou in Burkina Faso, Kersa and Addis Ababa in Ethiopia, and Ibadan and Lagos in Nigeria. In each country, one rural area (Nouna, Kersa, and a rural subarea in Ibadan) and one urban area (Ouagadougou, Addis Ababa, and Lagos) were included. The details of the study design and study population in Round 1 were described previously [43]. In the currently reported new round (Round 2), a rural subarea in Kintampo (Ghana), a rural subarea in Dodoma (Tanzania), and an urban subarea in Dar es Salaam (Tanzania) were added as new participating areas. Therefore, the Round 2 survey included nine areas from five countries. Although Ibadan and Dodoma are generally considered urban, the surveys were conducted in more rural subareas of Ibadan and Dodoma.

The detailed study protocol of the survey is available on the website of the Harvard University Center for African Studies (https://africa.harvard.edu/files/african-studies/files/arise_

covid_survey_round_2_methods_brief_final.pdf). Briefly, households were selected from sampling frames of existing Health and Demographic Surveillance Systems (HDSSs) or national surveys, where possible. Within the sampling frame in each area, we interviewed approximately 300 randomly selected households with adolescents between the ages of 10 to 19 years residing in the household. The study countries and study sites were selected based on research team capacity and infrastructure from existing collaborations in the African Research, Implementation Science and Education (ARISE) Network [44]. The target sample size was determined based on budget constraints, and we did not conduct a formal sample size calculation to determine the target sample size.

The methodology of the computer-assisted telephone interviewing was reported in detail previously [43]. Briefly, data collectors called each selected household to obtain consent to survey the household head or another adult member in the household. The data collectors then obtained parental or guardian consent to speak to the adolescent through the adult household member's phone if an adolescent was present in the household. If the adolescent was not present in the household (e.g., the adolescent was in school), an arrangement was made to call back at a later time. When contact was made with the adolescents through the parent's phone, the data collectors obtained assent directly from the adolescents and completed the adolescent interview.

The sampling frames and sampling methods for Burkina Faso, Ethiopia, and Nigeria have been described in detail previously [43]. For the new study area of Kintampo (Ghana), we used the Kintampo HDSS as the sampling frame. The Kintampo HDSS represents a catchment area of 163182 individuals and 39134 households. A total of 3589 adolescents were randomly sampled from the overall list of households, and 1074 households were called until the target sample size of 300 adolescents was reached. For the new study area of Dodoma (Tanzania), we used the Dodoma HDSS as the sampling frame; 600 households with adolescents were selected randomly from the sampling frame, and 318 eligible adolescents completed the survey. For the new study area of Dar es Salaam (Tanzania), we used the Dar es Salaam HDSS, also referred to as the Dar es Salaam Urban Cohort Study (DUCS) [45], as the sampling frame. The Dar es Salaam HDSS included 14754 households comprised of 143452 individuals, of which 30446 were adolescents. We randomly selected 2500 households, and approximately 655 households were contacted to attain the sample size of 302 adolescents.

The survey was conducted between July and December 2021. All households with adolescents aged 10 to 19 years that participated in the Round 1 survey (conducted between July and November 2020) were contacted again. For households that could not be contacted, new households were randomly identified from the sampling frame until the minimum target sample size of 300 households per area was reached. For Tanzania and Ghana, however, none of the adolescents participated in Round 1, as the two countries were added during the Round 2 survey.

## Data collection

The survey for adolescents included questions on adolescents' awareness, knowledge, perceptions, and attitudes toward COVID-19 vaccines, willingness to receive COVID-19 vaccines, potential determinants of COVID-19 vaccine hesitancy, trusted sources of information regarding COVID-19 vaccines, and the expectations on COVID-19 vaccine campaigns. The complete adolescent instrument used for the survey is available on the website of the Harvard University Center for African Studies (https://africa.harvard.edu/files/african-studies/files/arise_covid-19_survey_round_2_adolescent_household_survey_questionnaire.pdf).

### Statistical analysis

Data collection and statistical analysis were informed by the theoretical underpinning of the health belief model (Fig 1). Based on this conceptual framework, COVID-19 vaccine hesitancy among adolescents may be driven by perceived susceptibility to COVID-19 exposure, perceived severity of COVID-19 infection, perceived effectiveness and safety of COVID-19 vaccines, and cues to action received from media and other influencers (e.g., family members, teachers, healthcare providers). Several individual-level sociodemographic characteristics may also drive willingness to receive COVID-19 vaccines.

We conducted descriptive analyses on the sociodemographic characteristics of the adolescents, willingness to receive the COVID-19 vaccine, perceptions of the vaccine, self-reported determinants of their willingness to be vaccinated, and their trusted information sources and expectations about the vaccine. For the descriptive analyses, we calculated means and standard deviations (SDs) overall and by area for continuous variables and counts and percentages for categorical variables.

The outcome for the associational analyses was COVID-19 vaccine hesitancy, defined as a response of definitely not getting the COVID-19 vaccine or maybe, unsure, or undecided on whether to get the COVID-19 vaccine if it were available now. We defined vaccine hesitancy as a dichotomous outcome. The potential determinants included age, sex, country, rural residence, currently receiving education, perceived safety of the COVID-19 vaccine among children and adolescents, perceived effectiveness of the COVID-19 vaccine, current impacts of the COVID-19 pandemic on daily activities, levels of psychological distress measured using the four-item Patient Health Questionnaire for Depression and Anxiety Scale [46], perceived risk of being exposed to COVID-19, and knowledge score on the symptoms, transmission, and prevention of COVID-19 measured.

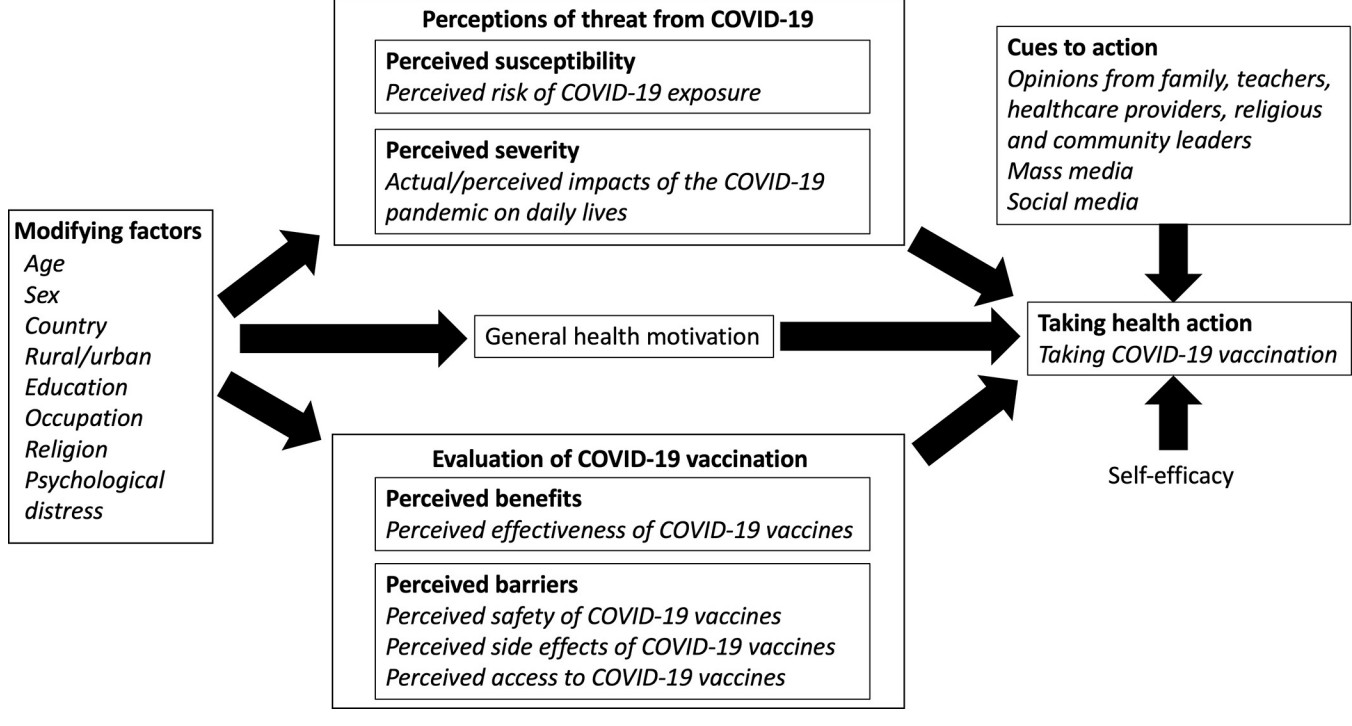

**Fig 1. Conceptual framework for COVID-19 hesitancy among adolescents based on the health belief model.**

We used log-binomial models [47, 48] to calculate the prevalence ratios (PRs) and 95% confidence intervals (CIs) and modified Poisson models to achieve model convergence whenever necessary [49]. We examined the crude associations between potential determinants and vaccine hesitancy in unadjusted models. We estimated adjusted prevalence ratios (aPRs) in adjusted models controlled for age, sex, country, and rural residence. We conducted all analyses using SAS 9.4 (SAS Institute Inc., Cary, North Carolina) at a two-sided $\alpha$ level of 0.05. Missing data were handled using the complete case analysis.

## Results

### Sociodemographic characteristics

A total of 2662 adolescents were interviewed (**Table 1**). The age of the adolescents ranged from a minimum of 10 years to a maximum of 19 years. The mean age ranged from 14 years in Dodoma (SD: 2.3) and Dar es Salaam (SD: 2.6) to 17 years in Addis Ababa (SD: 2.2). The sex distributions were roughly balanced between girls and boys, except for Kersa, where 33% were

**Table 1. Sociodemographic characteristics of adolescents in a phone-based survey in five sub-Saharan African countries, 2021[1].**

| | Burkina Faso | | Ethiopia | | Ghana | Nigeria | | Tanzania | | Total |
|---|---|---|---|---|---|---|---|---|---|---|
| | Rural | Urban | Rural | Urban | Rural | Rural | Urban | Rural | Urban | |
| | Nouna | Ouagadougou | Kersa | Addis Ababa | Kintampo | Ibadan | Lagos | Dodoma | Dar es Salaam | |
| Number of adolescents, N | 309 | 281 | 274 | 268 | 300 | 278 | 332 | 318 | 302 | 2662 |
| Adolescents who also participated in Round 1 survey, N (%) | 185 (59.9) | 162 (57.7) | 83 (30.3) | 170 (63.4) | 0 (0.0) | 91 (32.7) | 21 (6.3) | 0 (0.0) | 0 (0.0) | 712 (26.8) |
| Age, years (SD) | 15.8 (2.5) | 15.5 (2.4) | 15.7 (2.6) | 16.6 (2.2) | 15.6 (2.7) | 16.4 (1.8) | 15.8 (1.8) | 14.4 (2.3) | 14.1 (2.6) | 15.5 (2.5) |
| Girls, N (%) | 140 (45.3) | 150 (53.4) | 91 (33.2) | 172 (64.2) | 153 (51.0) | 152 (54.7) | 195 (58.7) | 175 (55.0) | 177 (58.6) | 1405 (52.8) |
| Highest level of education,[2,3] N (%) | | | | | | | | | | |
| None/religious school/literacy class | 38 (12.4) | 14 (5.0) | 2 (0.7) | 1 (0.4) | 10 (3.3) | 1 (0.4) | 0 (0.0) | 34 (10.7) | 0 (0.0) | 100 (3.8) |
| Some primary school | 90 (29.4) | 56 (19.9) | 136 (49.6) | 59 (22.0) | 109 (36.3) | 2 (0.7) | 0 (0.0) | 169 (53.1) | 109 (36.5) | 730 (27.5) |
| Completed primary school | 40 (13.1) | 16 (5.7) | 29 (10.6) | 26 (9.7) | 46 (15.3) | 8 (2.9) | 17 (5.1) | 65 (20.4) | 52 (17.4) | 299 (11.3) |
| Some secondary/high school | 114 (37.3) | 176 (62.6) | 92 (33.6) | 123 (45.9) | 127 (42.3) | 203 (73.3) | 294 (88.6) | 48 (15.1) | 113 (37.8) | 1290 (48.6) |
| Completed secondary/high school | 11 (3.6) | 17 (6.1) | 6 (2.2) | 40 (14.9) | 7 (2.3) | 58 (20.9) | 18 (5.4) | 2 (0.6) | 19 (6.4) | 178 (6.7) |
| Tertiary education or higher | 13 (4.3) | 2 (0.7) | 9 (3.3) | 19 (7.1) | 1 (0.3) | 5 (1.8) | 3 (0.9) | 0 (0.0) | 6 (2.0) | 58 (2.2) |
| Occupation[4] N (%) | | | | | | | | | | |
| Unemployed | 21 (6.8) | 11 (3.9) | 1 (0.4) | 3 (1.1) | 11 (3.7) | 0 (0.0) | 0 (0.0) | 71 (22.3) | 21 (7.0) | 139 (5.2) |
| Student | 161 (52.1) | 243 (86.5) | 256 (93.4) | 257 (95.9) | 259 (86.3) | 273 (98.9) | 332 (100.0) | 181 (56.9) | 267 (88.4) | 2229 (83.8) |
| Farmer | 86 (27.8) | 0 (0.0) | 13 (4.7) | 2 (0.8) | 14 (4.7) | 0 (0.0) | 0 (0.0) | 34 (10.7) | 1 (0.3) | 150 (5.6) |
| Wage employment | 0 (0.0) | 2 (0.7) | 0 (0.0) | 4 (1.5) | 3 (1.0) | 0 (0.0) | 0 (0.0) | 0 (0.0) | 3 (1.0) | 12 (0.5) |
| Self-employed | 7 (2.3) | 9 (3.2) | 2 (0.7) | 2 (0.8) | 8 (2.7) | 2 (0.7) | 0 (0.0) | 1 (0.3) | 10 (3.3) | 41 (1.5) |
| Casual, off-farm income | 6 (1.9) | 4 (1.4) | 0 (0.0) | 0 (0.0) | 1 (0.3) | 1 (0.4) | 0 (0.0) | 31 (9.8) | 0 (0.0) | 43 (1.6) |

[1] Values are mean (standard deviation) for age and count (percentage) for other variables. SD, standard deviation.

[2] Percentages may not add up to 100% due to rounding.

[3] Level of education missing for 3 adolescents in Nouna, 1 adolescent in Ibadan, and 3 adolescents in Dar es Salaam.

[4] Occupation missing for 2 adolescents in Ibadan.

girls, and Addis Ababa, where 64% were girls. The percentage of adolescents with at least some secondary school education ranged from 16% in Dodoma to 96% in Ibadan. Over 85% of the participants self-identified as students, except in Nouna and Dodoma, where 52% and 57% of the adolescents were students, respectively.

## Willingness to receive COVID-19 vaccines

Over 80% of adolescents had previously heard of COVID-19 vaccines, except in Nouna and Kersa, where 74% and 55% of the adolescents had heard of the vaccines, respectively (**Table 2**). Kersa had the highest percentage of adolescents who would definitely get vaccinated if a COVID-19 vaccine were available (86%), whereas Dodoma had the lowest percentage of willingness to definitely get vaccinated (12%). Correspondingly, Dodoma had the highest percentage of adolescents who would definitely *not* get vaccinated (79%), and Kersa had the lowest percentage (10%). Nouna had the highest percentage of adolescents unsure or undecided on whether they would be vaccinated if a COVID-19 vaccine were available (17%). In ascending order, the percentage of adolescents with COVID-19 vaccine hesitancy, defined as definitely not getting vaccinated or unsure/undecided, was 14% in Kersa, 23% in Ibadan, 31% in Nouna, 32% in Ouagadougou, 37% in Addis Ababa, 48% in Kintampo, 65% in Lagos, 76% in Dar es Salaam, and 88% in Dodoma. Analysis stratified by every single age and age bracket (young adolescents aged 10 to 14 years versus older adolescents aged 15 to 19 years) did not find consistent trends in COVID-19 vaccine hesitancy by age (**S1 Table**).

Among adolescents who were definitely or possibly willing to receive COVID-19 vaccines, keeping themselves and their families safe was the most common reason for vaccination, cited by over 92% of adolescents in all areas except in Dodoma (63%) and Dar es Salaam (89%). Parental or familial will was the second common driver for vaccination, selected by over 50% of adolescents in all areas except Addis Ababa (46%) and Dodoma (25%). Doctors' suggestions were the third common motivation for vaccination, chosen by over 40% of adolescents in all areas except Dodoma (16%) and Dar es Salaam (35%). Among adolescents who were definitely not or possibly not willing to receive COVID-19 vaccines, the leading reasons for hesitancy were perceived low necessity (selected by 47% of adolescents overall), concerns about the safety of the vaccine (selected by 45% overall), and concerns about the effectiveness of the vaccine (selected by 11% overall). Personal liberty concerns, the influence of conspiracy theories, religious reasons, and fears of getting unlicensed, experimental, or worse-quality vaccines were not the main reasons for hesitancy, each cited by less than 3% of adolescents.

## Perceptions of COVID-19 vaccines

The percentage of adolescents who perceived COVID-19 vaccines to be very safe or somewhat safe among the general population ranged from 38% in Dodoma to 77% in Addis Ababa (**S2 Table**). Similarly, the percentage of adolescents perceiving COVID-19 vaccines as very safe or somewhat safe among children and adolescents varied from 33% in Dodoma to 77% in Kersa. In terms of the perceived effectiveness of COVID-19 vaccines among the general population, 39% of participants in Dodoma to 80% of participants in Kersa perceived COVID-19 vaccines to be very effective or somewhat effective in preventing COVID-19 infection. Overall, 46% of adolescents could not identify any possible side effects of COVID-19 vaccines, and 27% believed there were no side effects.

## Self-reported determinants of willingness to receive vaccines

The percentage of adolescents whose willingness to receive COVID-19 vaccines would be affected by the vaccine's country of origin ranged from 15% in Ibadan to 45% in Lagos

**Table 2. Willingness to receive COVID-19 vaccines among adolescents in a phone-based survey in five sub-Saharan African countries, 2021[1].**

| | Burkina Faso | | Ethiopia | | Ghana | Nigeria | | Tanzania | | Total |
|---|---|---|---|---|---|---|---|---|---|---|
| | Rural | Urban | Rural | Urban | Rural | Rural | Urban | Rural | Urban | |
| | Nouna | Ouagadougou | Kersa | Addis Ababa | Kintampo | Ibadan | Lagos | Dodoma | Dar es Salaam | |
| Number of adolescents, $N$ | 309 | 281 | 274 | 268 | 300 | 278 | 332 | 318 | 302 | 2662 |
| Having heard of COVID-19 vaccine,[2] $N$ (%) | 225 (73.5) | 251 (89.3) | 146 (55.3) | 263 (98.1) | 261 (87.9) | 240 (88.6) | 323 (97.3) | 267 (84.0) | 295 (98.0) | 2271 (86.1) |
| Willingness to receive COVID-19 vaccine if it were available now,[3,4] $N$ (%) | | | | | | | | | | |
| Would definitely not get vaccinated | 30 (13.3) | 64 (25.6) | 15 (10.3) | 77 (29.3) | 92 (35.3) | 36 (15.0) | 172 (53.3) | 210 (78.7) | 198 (67.1) | 894 (39.4) |
| Would definitely get vaccinated | 156 (69.3) | 171 (68.4) | 126 (86.3) | 166 (63.1) | 136 (52.1) | 186 (77.5) | 114 (35.3) | 32 (12.0) | 72 (24.4) | 1159 (51.1) |
| Maybe/unsure/undecided | 39 (17.3) | 15 (6.0) | 5 (3.4) | 20 (7.6) | 33 (12.6) | 18 (7.5) | 37 (11.5) | 25 (9.4) | 25 (8.5) | 217 (9.6) |
| Reasons for getting the COVID-19 vaccine,[5,6] $N$ (%) | | | | | | | | | | |
| To keep self and family safe | 182 (93.3) | 183 (98.4) | 127 (97.0) | 184 (98.9) | 156 (92.3) | 198 (97.1) | 145 (96.0) | 36 (63.2) | 86 (88.7) | 1297 (94.3) |
| Parents' or family's will | 172 (88.2) | 163 (87.6) | 97 (74.1) | 86 (46.2) | 125 (74.0) | 171 (83.8) | 89 (58.9) | 14 (24.6) | 69 (71.1) | 986 (71.7) |
| Because doctor suggested | 156 (80.0) | 180 (96.8) | 62 (47.3) | 84 (45.2) | 119 (70.4) | 154 (75.5) | 70 (46.4) | 9 (15.8) | 34 (35.1) | 868 (63.1) |
| Reasons for not getting the COVID-19 vaccine,[5,7] $N$ (%) | | | | | | | | | | |
| Perceived low necessity of being vaccinated | 48 (71.6) | 45 (60.0) | 13 (65.0) | 31 (32.3) | 20 (18.0) | 18 (36.0) | 79 (38.2) | 168 (90.8) | 59 (26.8) | 481 (46.7) |
| Concerns about effectiveness of the vaccine | 9 (13.4) | 24 (32.0) | 0 (0.0) | 20 (20.8) | 7 (6.3) | 5 (10.0) | 40 (19.3) | 2 (1.1) | 8 (3.6) | 115 (11.2) |
| Concerns about safety of the vaccine | 24 (35.8) | 49 (65.3) | 8 (40.0) | 64 (66.7) | 59 (53.2) | 24 (48.0) | 142 (68.6) | 6 (3.2) | 88 (40.0) | 464 (45.0) |
| Fear of getting unlicensed/experimental/worse-quality vaccines | 2 (3.0) | 1 (1.3) | 0 (0.0) | 2 (2.1) | 3 (2.7) | 1 (2.0) | 10 (4.8) | 1 (0.5) | 2 (0.9) | 22 (2.1) |
| Religious reasons | 0 (0.0) | 0 (0.0) | 0 (0.0) | 11 (11.5) | 1 (0.9) | 0 (0.0) | 2 (1.0) | 0 (0.0) | 1 (0.5) | 15 (1.5) |
| Influence of conspiracy theories | 1 (1.5) | 1 (1.3) | 0 (0.0) | 4 (4.2) | 13 (11.7) | 2 (4.0) | 4 (1.9) | 0 (0.0) | 0 (0.0) | 25 (2.4) |
| Personal liberty concerns | 0 (0.0) | 2 (2.7) | 0 (0.0) | 2 (2.1) | 5 (4.5) | 0 (0.0) | 1 (0.5) | 2 (1.1) | 3 (1.4) | 15 (1.5) |

[1] Values are counts (percentages) for categorical variables.

[2] Missing for 3 adolescents in Nouna, 10 adolescents in Kersa, 7 adolescents in Ibadan, 1 adolescent in Dar es Salaam, and 3 adolescents in Kintampo.

[3] Percentages may not add up to 100% due to rounding.

[4] Among adolescents who have heard of COVID-19 vaccines. Missing for 1 adolescent in Ouagadougou.

[5] Counts and percentages do not add up to the total because the selection of multiple reasons was allowed.

[6] Counts and percentages are among adolescents who would definitely get vaccinated and who were unsure/undecided.

[7] Counts and percentages are among adolescents who would definitely not get vaccinated and who were unsure/undecided. Missing for 2 adolescents in Nouna, 4 adolescents in Ouagadougou, 1 adolescent in Addis Ababa, 4 adolescents in Ibadan, 2 adolescents in Lagos, 3 adolescents in Dar es Salaam, 50 adolescents in Dodoma, and 14 adolescents in Kintampo.

(**S3 Table**). In descending order, the overall percentages of adolescents willing to receive COVID-19 vaccines developed by specific countries were 36% for the United States, 22% for China, 15% for Europe, 14% for Russia, and 11% for India. Overall, 30% of the participants would be more willing to receive a COVID-19 vaccine developed or tested in Africa, ranging from 15% in Ibadan to 48% in Kintampo and Ouagadougou. The individuals or groups that

had the greatest impacts on the adolescents' willingness to receive COVID-19 vaccines were healthcare workers (60% overall), parents or family members (58% overall), and schoolteachers (46% overall). Religious leaders affected COVID-19 vaccine willingness for over 20% of adolescents in all areas (36% overall).

### Trusted information sources and expectations of vaccines

The most highly trusted sources of information on COVID-19 vaccines for adolescents were television, radio, or newspaper (85% overall), medical professionals (83% overall), and governmental communications (73% overall) (S4 Table). The adolescents' willingness to participate in a COVID-19 vaccine trial ranged from 14% in Dodoma to 66% in Kersa. Less than 2% of the adolescents had already received a COVID-19 vaccine, except in Ibadan, where 13% had been vaccinated. Overall, 41% of the adolescents did not know when a COVID-19 vaccine would be made available to them; 70% of adolescents in Dodoma and 41% in Dar es Salaam thought that a COVID-19 vaccine would never be made available to them.

### Potential determinants of COVID-19 vaccine hesitancy

Compared to girls, boys had an 8% lower prevalence of vaccine hesitancy (aPR: 0.92; 95% CI: 0.86, 0.99) (Table 3). Compared to adolescents in Burkina Faso, adolescents in Ethiopia had a

**Table 3. Potential determinants of COVID-19 vaccine hesitancy among adolescents in a phone-based survey in five sub-Saharan African countries, 2021[1].**

| | COVID-19 vaccine hesitancy[2] | |
| --- | --- | --- |
| | **Unadjusted** | **Adjusted** |
| | PR (95% CI) | aPR (95% CI) |
| Age, years | 0.95 (0.93, 0.97) | 0.98 (0.97, 1.00) |
| Male sex | 0.86 (0.79, 0.94) | 0.92 (0.86, 0.99) |
| Country | | |
| Burkina Faso | 1.00 (Ref) | 1.00 (Ref) |
| Ethiopia | 0.92 (0.75, 1.13) | 0.92 (0.75, 1.13) |
| Ghana | 1.54 (1.28, 1.85) | 1.59 (1.32, 1.92) |
| Nigeria | 1.50 (1.28, 1.76) | 1.51 (1.29, 1.78) |
| Tanzania | 2.62 (2.28, 3.01) | 2.51 (2.18, 2.89) |
| Rural residence | 0.82 (0.75, 0.89) | 0.93 (0.86, 1.01) |
| Not currently receiving education | 0.81 (0.70, 0.93) | 1.06 (0.93, 1.21) |
| Perceived safety of the COVID-19 vaccine among children and adolescents | | |
| Very safe | 1.00 (Ref) | 1.00 (Ref) |
| Somewhat safe | 1.97 (1.65, 2.37) | 1.87 (1.57, 2.22) |
| Not very safe | 3.91 (3.35, 4.57) | 3.52 (3.02, 4.10) |
| Not safe at all | 4.59 (3.94, 5.35) | 3.52 (3.00, 4.13) |
| Do not know | 3.35 (2.85, 3.93) | 2.83 (2.42, 3.32) |
| Perceived effectiveness of the COVID-19 vaccine | | |
| Very effective | 1.00 (Ref) | 1.00 (Ref) |
| Somewhat effective | 2.08 (1.76, 2.47) | 1.94 (1.65, 2.28) |
| Not very effective | 3.86 (3.32, 4.48) | 3.29 (2.83, 3.81) |
| Not effective at all | 4.60 (3.99, 5.31) | 3.46 (2.97, 4.03) |
| Do not know | 3.29 (2.82, 3.82) | 2.77 (2.38, 3.22) |
| Current impacts of the COVID-19 pandemic on daily activities | | |
| No impacts | 1.00 (Ref) | 1.00 (Ref) |

*(Continued)*

**Table 3.** (Continued)

| | COVID-19 vaccine hesitancy[2] | |
| --- | --- | --- |
| | **Unadjusted** | **Adjusted** |
| Some impacts | 1.18 (1.08, 1.29) | 0.89 (0.81, 0.98) |
| Psychological distress[3] | | |
| No psychological distress | 1.00 (Ref) | 1.00 (Ref) |
| Mild psychological distress | 0.99 (0.87, 1.12) | 1.05 (0.95, 1.16) |
| Moderate psychological distress | 0.91 (0.72, 1.15) | 0.99 (0.81, 1.21) |
| Severe psychological distress | 0.89 (0.51, 1.55) | 0.93 (0.60, 1.43) |
| Perceived risk of exposure to COVID-19[4] | | |
| No risk | 1.00 (Ref) | 1.00 (Ref) |
| Low risk | 0.93 (0.66, 1.30) | 0.93 (0.66, 1.30) |
| High risk | 0.75 (0.51, 1.11) | 0.86 (0.57, 1.30) |
| Very high risk | 0.76 (0.41, 1.43) | 0.81 (0.43, 1.53) |
| High knowledge score of COVID-19[5] | 0.95 (0.74, 1.22) | 0.83 (0.65, 1.07) |

[1] Values are prevalence ratios (95% confidence intervals) from log-binomial or modified Poisson models. The adjusted analyses controlled for age, sex, country, and rural residence. PR, prevalence ratio; CI, confidence interval; aPR, adjusted prevalence ratio.

[2] COVID-19 vaccine hesitancy was defined as a response of definitely not getting the COVID-19 vaccine, or a response of maybe, unsure, or undecided on whether to get the COVID-19 vaccine if it were available now.

[3] Psychological distress was measured using the four-item Patient Health Questionnaire for Depression and Anxiety Scale. Each item had a numeric score of 0, 1, 2, and 3, and the total score was computed by adding up the four items, resulting in a total score ranging from 0 to 12. The total score of psychological distress was categorized into none (total score: 0 to 2), mild (total score: 3 to 5), moderate (total score: 6 to 8), and severe (total score: 9 to 12).

[4] For adolescents in Burkina Faso, Ethiopia, and Nigeria who participated in the Round 1 survey.

[5] For adolescents in Burkina Faso, Ethiopia, and Nigeria who participated in the Round 1 survey. For each of the three domains of COVID-19 knowledge (symptoms, transmission, and prevention), a score was created based on the number of correct responses. The maximum scores for symptoms, transmission, and prevention were 10, 5, and 7, respectively. The three scores were then added to construct a total knowledge score of COVID-19 with a range from 0 to 22. A high knowledge score was defined as a total score of 18 or greater.

similar prevalence of COVID-19 vaccine hesitancy (aPR: 0.92; 95% CI: 0.75, 1.13), adolescents in Ghana had 59% higher prevalence of COVID-19 vaccine hesitancy (aPR: 1.59; 95% CI: 1.32, 1.92), adolescents in Nigeria had 51% higher prevalence of COVID-19 vaccine hesitancy (aPR: 1.51; 95% CI: 1.29, 1.78), and adolescents in Tanzania had 2.5 times the prevalence of COVID-19 vaccine hesitancy (aPR: 2.51; 95% CI: 2.18, 2.89). Compared to those who perceived COVID-19 vaccines to be very safe among children and adolescents, adolescents perceiving COVID-19 vaccines to be somewhat safe, not very safe, and not safe at all had 1.9 times (aPR: 1.87; 95% CI: 1.57, 2.22), 3.5 times (aPR: 3.52; 95% CI: 3.02, 4.10), and 3.5 times (aPR: 3.52; 95% CI: 3.00, 4.13) the prevalence of COVID-19 vaccine hesitancy, respectively. Compared to adolescents who believed that COVID-19 vaccines were very effective, adolescents who perceived COVID-19 vaccines to be somewhat effective, not very effective, and not effective at all had 1.9 times (aPR: 1.94; 95% CI: 1.65, 2.28), 3.3 times (aPR: 3.29; 95% CI: 2.83, 3.81), and 3.5 times (aPR: 3.46; 95% CI: 2.97, 4.03) the prevalence of vaccine hesitancy, respectively. Experiencing any current impacts of the COVID-19 pandemic on daily activities was associated with an 11% lower prevalence of COVID-19 vaccine hesitancy (aPR: 0.89; 95% CI: 0.81, 0.98).

## Discussion

This survey across five sub-Saharan African countries assessed the prevalence of COVID-19 vaccine hesitancy in adolescents and identified the potential determinants of adolescent COVID-19 vaccine hesitancy. We find high COVID-19 vaccine hesitancy among adolescents from the nine areas across five countries in sub-Saharan Africa. The prevalence of hesitancy is especially concerning in Tanzania. Individual characteristics that potentially increase vaccine hesitancy are female sex, perceived lack of safety, and perceived lack of effectiveness of COVID-19 vaccines.

COVID-19 vaccine perceptions and hesitancy are associated with social and cultural factors that vary considerably across settings [14–20]. Of note, a large sample of 13,426 participants from the general population in 19 countries shows high heterogeneity across countries in the determinants, and even the directions of associations, of vaccine acceptance [15]. Our study is, to our knowledge, the first study that examined the prevalence and determinants of COVID-19 vaccine hesitancy among adolescents across various settings in sub-Saharan Africa. While the proportion of adolescents with COVID-19 vaccine hesitancy is high across areas, considerable variation is also noted, ranging from 14% in Kersa to 88% in Dodoma. Previous studies in high- or upper-middle-income settings report similarly heterogeneous prevalences of hesitancy. A study in adolescents aged 12 to 15 years in Arkansas, United States, reports that 58% are hesitant about getting a COVID-19 vaccine [21]. A study among Canadian adolescents aged 14 to 17 years reports that 35% would not get a COVID-19 vaccine or were unsure [23]. Based on a survey conducted by the United States CDC in April 2021, only 52% of unvaccinated adolescents aged 13 to 17 years would definitely or probably receive COVID-19 vaccination [11]. A study among Swedish adolescents shows that nearly one in three adolescents had not decided if they wanted to get a COVID-19 vaccine and that only 54% were willing to be vaccinated [24]. In a survey among Chinese adolescents, 76% of the participants would accept future COVID-19 vaccination [9]. In our survey, the exceptionally high prevalence of vaccine hesitancy in Dodoma and Dar es Salaam is perhaps attributable to the misconceptions regarding COVID-19 and its vaccination, partially exacerbated by misinformation among communities in 2020 and early 2021 [50, 51].

We find that healthcare providers are adolescents' most trusted individuals for information on COVID-19 vaccines and have sizable impacts on the adolescents' willingness to vaccinate, even in Tanzania, where vaccine hesitancy is high. Previous global surveys among the general population consistently show that health workers are the most trusted sources of guidance about COVID-19 vaccines [29]. Among Turkish adults, higher trust in health professionals is associated with more favorable attitudes towards COVID-19 vaccines [42]. Our survey also suggests that parents', friends', and schoolteachers' attitudes toward COVID-19 vaccination may be associated with adolescents' vaccine willingness. Similarly, a sample of adolescents in the United States shows that parent and peer norms are predictors of adolescent willingness to receive vaccines [22]. Therefore, healthcare workers, parents, schoolteachers, and peers should be leveraged as advocates of COVID-19 vaccines, and efforts are needed to increase vaccine acceptance among healthcare workers and adult community members.

Our survey finds that, in all study areas, over 40% of adolescents consider religious bodies or leaders a trusted source of information on COVID-19 vaccines and that religious leaders affect COVID-19 vaccine willingness for over 20% of adolescents. There is mixed evidence regarding religious influence on COVID-19 vaccine perceptions. A survey among Turkish adults finds no significant associations between religious attitudes and COVID-19 vaccine perceptions [33]. However, a survey in Pakistan shows a strong association between religious inhibitions (the belief that trust in God was sufficient to protect one from infection) and vaccine

hesitancy in adults [35]. In our survey, for 15% of adolescents overall, vaccine willingness can be affected by celebrities and social media influencers. The uncritical consumption of social media may promote vaccine hesitancy [52]. In particular, emerging social media platforms may be used to spread anti-vaccination content that jeopardizes vaccine uptake, particularly for young people who are more likely to use such platforms [53]. Social media may also spread misinformation or conspiracy theories about COVID-19. In a survey among adults in Jordan, participants who believe in COVID-19 conspiracy theories are less likely to accept COVID-19 vaccines [36]. Future efforts are needed to leverage religious leaders and social media as potential mechanisms for promoting adolescents' uptake of COVID-19 vaccines.

We report that the vaccine willingness of 15% or more of adolescents in all areas is affected by the vaccine's country of origin, with COVID-19 vaccines developed in the United States the most accepted, followed by those developed in China; vaccines developed in Russia and India appear the least accepted. Among adults in Kazakhstan [34] and Zimbabwe [25], vaccine hesitancy is likewise associated with the vaccines' country of origin or manufacturer. In Kazakhstan, for instance, 78% of the respondents have the highest confidence in German-produced vaccines and the lowest confidence in vaccines produced in India [34]. A global survey of 19,714 adults from 17 countries shows that 52% of the respondents would only accept COVID-19 vaccines from a specific country of origin [19]. In our survey, over 15% of the adolescents in all areas are more willing to receive COVID-19 vaccines developed or tested in Africa, highlighting the importance of technology transfer for COVID-19 vaccines to increase sub-Saharan African countries' local capacities.

Concerns about the safety and effectiveness of COVID-19 vaccines are among the top reasons for hesitancy in this survey. Perceived lack of safety and perceived lack of effectiveness strongly predict greater vaccine hesitancy. These findings align with previous studies among the adult population in both high-income and low- and middle-income settings [19, 29, 54]. A multi-country study in the United States, Russia, and 10 LMICs in Asia, Africa, and South America [29] finds that vaccine acceptance in LMICs is primarily driven by the desire for personal protection, and that the most common reason for hesitancy is the concern about the side effects of the vaccine [29]. Among Chinese adolescents, those who think that COVID-19 vaccines can protect them from COVID-19 infection and those who believe that vaccines are safe are more likely to be willing to receive COVID-19 vaccines [9]. Therefore, public health campaigns for adolescents and their guardians to boost COVID-19 vaccine uptake should emphasize that the vaccines are safe for adolescents and confer protection against COVID-19 infections [10].

We find that adolescent boys have less COVID-19 vaccine hesitancy than adolescent girls, consistent with existing evidence. In Turkish adults, males are more likely to have positive attitudes toward COVID-19 vaccines [33]. Similarly, in a mixed-methods study in Sweden, girls have higher anxiety levels about the vaccine than boys. The specific mechanisms for the greater vaccine hesitancy among adolescent girls are unclear but may be related to the lower tendency of risk-taking among adolescent girls than among boys (i.e., considering receiving COVID-19 vaccines as a risk-taking behavior) [55]. These findings highlight the need to address COVID-19 vaccine hesitancy, especially among adolescent girls. A study among 472,521 adults in Latin America and the Caribbean finds that having depressive symptoms is associated with greater fear of the adverse effects of the COVID-19 vaccines [39], indicating that vaccine hesitancy for some individuals may be related to underlying mental health issues that must be addressed accordingly. We do not find associations between psychological distress and COVID-19 vaccine hesitancy. The impacts of the COVID-19 pandemic on the long-term mental well-being of sub-Saharan African adolescents need to be investigated further.

The strengths of this study are the inclusion of multiple countries and cities in sub-Saharan Africa and the measurement of a wide array of potential determinants of COVID-19 vaccine hesitancy. A potential limitation of the study is that the study areas were selected based on existing connections and infrastructure, and the adolescents in each area were not selected probabilistically. Therefore, the results from this study may not be generalized to all adolescents in the country. Nevertheless, we increased the representativeness of the study population by including countries and areas geographically spread across sub-Saharan Africa. Therefore, the results provide important insights into the prevalence of determinants of COVID-19 vaccine hesitancy in sub-Saharan Africa. As another limitation, due to the nature of phone-based interviews, the adolescents included in most areas were those who resided in households with access to a mobile phone. Consequently, the sample might underrepresent adolescents from under-resourced households, which may affect generalizability.

In conclusion, we show that COVID-19 vaccine hesitancy among adolescents is high across the five sub-Saharan African countries, especially in Tanzania. COVID-19 vaccination campaigns among sub-Saharan African adolescents must address adolescents' concerns and misconceptions about COVID-19 vaccines, especially regarding vaccination safety and effectiveness. It is crucial to ensure that vaccines are accessible should adolescents desire to be vaccinated. It rests upon the global medical community to get the shots into the arms of the often-neglected population of sub-Saharan African adolescents.

## Supporting information

**S1 Table. COVID-19 vaccine hesitancy by age among adolescents in a phone-based survey in five sub-Saharan African countries, 2021.**
(DOCX)

**S2 Table. Perceptions of the COVID-19 vaccine among adolescents in a phone-based survey in five sub-Saharan African countries, 2021.**
(DOCX)

**S3 Table. Determinants of the willingness to receive the COVID-19 vaccine among adolescents in a phone-based survey in five sub-Saharan African countries, 2021.**
(DOCX)

**S4 Table. Trusted information sources and expectations about the COVID-19 vaccine among adolescents in a phone-based survey in five sub-Saharan African countries, 2021.**
(DOCX)

## Acknowledgments

We thank the study participants and data collectors for making this study possible. The survey team in Ghana is grateful for support from the Kintampo Health Research Centre of Ghana Health Service and the community leadership of Kintampo North Municipality and Kintampo South District.

## Author Contributions

**Conceptualization:** Dongqing Wang, Mary Mwanyika-Sando, Till Baernighausen, Tajudeen Raji, Wafaie W. Fawzi.

**Data curation:** Angela Chukwu, Mary Mwanyika-Sando, Sulemana Watara Abubakari, Nega Assefa, Abbas Ismail, Bruno Lankoande, Frank Mapendo, Ourohiré Millogo, Firehiwot

Workneh, Temesgen Azemraw, Lawrence Gyabaa Febir, Christabel James, Amani Tinkasimile, Kwaku Poku Asante, Yemane Berhane, Japhet Killewo, Ayoade M. J. Oduola, Ali Sie, Abdramane Bassiahi Soura, Said Vuai.

**Formal analysis:** Dongqing Wang.

**Funding acquisition:** Till Baernighausen, Emily R. Smith, Wafaie W. Fawzi.

**Methodology:** Dongqing Wang, Angela Chukwu, Mary Mwanyika-Sando, Isabel Madzorera, Elena C. Hemler, Emily R. Smith.

**Project administration:** Elena C. Hemler, Wafaie W. Fawzi.

**Supervision:** Said Vuai, Wafaie W. Fawzi.

**Writing – original draft:** Dongqing Wang.

**Writing – review & editing:** Dongqing Wang, Angela Chukwu, Mary Mwanyika-Sando, Sulemana Watara Abubakari, Nega Assefa, Isabel Madzorera, Elena C. Hemler, Abbas Ismail, Bruno Lankoande, Frank Mapendo, Ourohiré Millogo, Firehiwot Workneh, Temesgen Azemraw, Lawrence Gyabaa Febir, Christabel James, Amani Tinkasimile, Kwaku Poku Asante, Till Baernighausen, Yemane Berhane, Japhet Killewo, Ayoade M. J. Oduola, Ali Sie, Emily R. Smith, Abdramane Bassiahi Soura, Tajudeen Raji, Said Vuai, Wafaie W. Fawzi.

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
