## [Decision Letter · Decision Letter 0]

11 Jul 2022

PGPH-D-22-00807

Levels and determinants of COVID-19 vaccine hesitancy among sub-Saharan African adolescents

Dear Dr. Wang,

Thank you for submitting your manuscript to PLOS Global Public Health. After careful consideration, we feel that it has merit but does not fully meet PLOS Global Public Health’s publication criteria as it currently stands. Therefore, we invite you to submit a revised version of the manuscript that addresses the points raised during the review process.

EDITOR'S COMMENTS: 

The study reported in the manuscript is very important and suggestions for revision offered are intended to strengthen the thesis and theoretical underpinning of its argument to facilitate achieving high standard manuscript that will contribute significantly to the contemporary issue in global conversation regarding Covid-19 vaccine hesitancy among adolescents.The two reviewers have made their respective observations and both have covered the spectrum in the domain of the weaknesses of the manuscript, no doubt.Kindly address all issues pointed out, especially, to include a paragraph in the introduction section that provides theoretical and conceptual clarifications underpinning the problem dynamics. This will give the thesis the scientific foundation it requires.

We look forward to receiving your revised manuscript.

Kind regards,

Nnodimele Onuigbo Atulomah, PhD

Academic Editor

Journal Requirements:

1.  Please amend your detailed online Financial Disclosure statement. This is published with the article. It must therefore be completed in full sentences and contain the exact wording you wish to be published.

2. Please update your online Competing Interests statement. If you have no competing interests to declare, please state: “The authors have declared that no competing interests exist.”

Additional Editor Comments (if provided):

The two reviewers that reviewed the manuscript have made important recommendation based on their expertise and observations of what this manuscript requires to express the underlying thesis it seeks to argue. The study covers an important theme of contemporary public health issue appearing in learned journals which readers may find interesting to read and learn important lessons by the time corrections are effected and manuscript is published. We at PLoS Global Public Health are encouraged and eager to bring this work to the readership of our discerning academic community hence these suggested revisions are essential for this to happen.

1. The title appear good and addresses contemporary issue of wide impact.

2. The population of interest is broad covering a wide area of at-risk population, this is good.

3. The study is premised on behaviour diagnosis of the likelihood of accepting to receive COVID-19 vaccine based on predictive value of the relationship between set of predictors variables.

However there are equally observed weaknesses in not conceptualizing the study appropriately in theory-grounded behavioral science of health-seeking operationalized in the health belief model. The manuscript would be strengthened id this is addressed, since the word "determinants" and the variables in the health belief model includes knowledge, attitude and perception, makes it easy to include a paragraph of theoretical and conceptual clarification of how these variables may provide elucidation of the dynamics associated with the problem phenomenon. Remember, the health belief model is the theory underpinning the identified problem

phenomenon.

The study would not contribute much if it has not attempted to elucidation the dynamics involved in the findings from the data guided by the theory.

Other issues observed require minor attention and revisions.

Reviewers' comments:

Reviewer's Responses to Questions

**Comments to the Author**

1. Does this manuscript meet PLOS Global Public Health’s publication criteria? Is the manuscript technically sound, and do the data support the conclusions? The manuscript must describe methodologically and ethically rigorous research with conclusions that are appropriately drawn based on the data presented.

Reviewer #1: Yes

Reviewer #2: Yes

2. Has the statistical analysis been performed appropriately and rigorously?

Reviewer #1: Yes

Reviewer #2: Yes

3. Have the authors made all data underlying the findings in their manuscript fully available (please refer to the Data Availability Statement at the start of the manuscript PDF file)?

Reviewer #1: Yes

Reviewer #2: Yes

4. Is the manuscript presented in an intelligible fashion and written in standard English?

Reviewer #1: Yes

Reviewer #2: Yes

5. Review Comments to the Author

Reviewer #1: The title for the study appears appropriate, the study is important in so many ways but most significantly, it describes population preventive health issues of great concern, and it is premised on behaviour diagnosis of the likelihood of accepting to receive COVID-19 vaccine based on predictive value of the relationship between set of predictors including information resources which, to an extent, was left out and the outcome variable in the study.

This reviewer observed that in the theoretical conceptualization, the study has overlooked the value of establishing its thesis on adequate behavioural grounding by mere selection of two variables perception and attitudinal disposition without recourse to sound theoretical basis, for instance, how certain behaviour theories may be well positioned in elucidating the dynamics of the problem phenomenon- COVID-19 vaccine hesitancy among sub-Saharan African adolescents by applying the principles of health-seeking behaviour and it typologies operationalized by the health belief model which clearly identifies health knowledge gained from information adequacy from the environment among others. The absence of proper scientific foundation in conceptualizing this study has weakened its argument. The study lacked adequate conceptual framework to tailor the research questions, what the research is hypothesizing, the development of the instrument for data collection.

A. OBSERVATIONS: This study is such an important study which should hold important answers within its thesis and narrative. Inevitably, there are major issues to be addressed to strengthen the argument of the authors as to what constitute the determinants of COVID-19 vaccine hesitancy among sub-Saharan African adolescents and the levels –(in terms of measures of these variables and constructs and not merely frequency distribution counts) of the variables; more of synthesizing standardized scores to compare across the multi-country study.

i. The Abstract. The countries mentioned in lines 52-53, were study locations and participants selected on the basis of random balloting or by convenience sampling and how was 300 adolescent participants computed as sample size? Clarification on critical appraisal providing the choice of variables for considerations in selecting appropriate behaviour theories related to variable of interest, and the corresponding antecedent variables which would provide possible elucidation of the problem phenomenon. Line 57: Why not state logistic regression analysis instead of Log-binomial models. Is this a different statistical analysis?

ii. Introduction. The introduction has appropriately considered, for clarity of sequence in its content, contextual background, and scoping background (empirical review) but weakly glossing over the theoretical and conceptual clarification in lines 115-116 which is necessary for providing evidence for the needed understanding sought by the study to elucidate the dynamics of the problem phenomenon requiring the study. But the theoretical and conceptual resources required to achieve this has been underplayed in characterizing the science underpinning the problem phenomenon. Theoretical and conceptualization Measurement challenges. Observed lack of sound conceptualization of the problem phenomenon grounded in appropriate behaviour theories since hesitancy just as intention proxy behaviour outcomes has weaken the argument of determinant of Vaccine hesitancy among adolescents.

Where is the review of the theoretical foundation that should provide the explanations of the dynamics of the phenomenon? For this type of study that is contexed in risk of an adverse outcome should be based on the principles of health-seeking explainable from the framework of health belief and concept of prevention. The absence of this is a serious omission to establish the thesis of the study, which renders it superficial and weak.

In the section for data collection Line 178 – 186, there seems to be inconsistencies regarding variables in this study. There is mentioned knowledge in Line 179 as a variable measured along with perception and attitudinal disposition, but line 103 omits it and the result does not show a measure for knowledge. The apparent disconnect observed demonstrates the lack of adequate conceptualization and application of an appropriate conceptual framework.

These are not appropriately measured in this study and has weakened the presentation of the result and conclusions.

The statistical analysis is completely flawed by not developing weighted aggregate scores for these variables representing the personal-level chrematistics of the respondents in the various countries and standardizing the resultant means for comparison. This would have been simpler and easily comprehensible, than the complex tables of frequency distribution so difficult to answer the research questions.

The results as presented appear very busy. How would the study reconcile the contents of table 1, 2 and 3? It would have been simpler to identify if the sociodemographic characteristics of participants from the rural settings are matched and same for urban setting across countries. The explanatory variables would have been computed as aggregate scores for each country for both rural and urban settings and standardize the mean scores to determine where they are located on the distribution of means of standardized scores.

Discussion: It would have been very appropriate to reiterate the purpose of the study as an introduction for the discussion before presenting findings. It would be better to use the term "associated with" than use "influenced" in line 348, 404 and elsewhere because this cannot be determined by the study design implemented. Revise where the word “influence” has been used accordingly except in the context of likelihood.

Line 352: Note the use of the word “level” implies measurement requiring scale of measures involving scoring the variables constituting the determinants particularly if they are in the domain of interval scale or ratio scales.

Reviewer #2: PLOS GLOBAL PUBLIC HEALTH

LEVELS AND DETERMINANTS OF COVID-19 VACCINE HESITANCY AMONG SUB-SAHARAN AFRICAN ADOLESCENTS.

This article by Dongqing et al, focused on covid-19 vaccine hesitancy among adolescents in a cross-country survey to determine reasons for hesitancy and offer recommendations or solutions to the challenges encountered in the global efforts to control the covid-19 pandemic. This paper reported that the adolescents constitute about 23% of the population of sub-Saharan Africa. (line 88), with a mortality and morbidity from corona virus found to be lower than in adults. This paper opined that there was a strong rational for providing covid-19 vaccines to adolescents because infected adolescents could transmit the virus to other individuals as well as develop severe symptoms or complications hence the need for vaccination.

This article is very appropriate and timely as it addresses a fundamental global problem that has affected the human race with a view to providing a solution to the challenge posed by the corona virus pandemic in sub-Saharan Africa.

THE TITLE: The title is very appealing and appropriate. It would have been more explicit if the authors distinguished the words “levels “and “Determinants” in more concise terms with detailed analysis inside the article.

THE ABSTRACT: The abstract was able to summarize concisely the research that was done with in-depth display of the essential information required.

INTRODUCTION: The authors were able to give an over view of the pandemic, the different vaccines and why the adolescents needs vaccination.

METHODOLOGY: This article presented a unique study design, study population, data collection and statistical analysis. But there are some points that will require further clarifications:

1. The authors reported this article as “an ongoing survey that used a novel mobile platform and computer-assisted telephone interviewing to collect data from Sub-Saharan African adolescents, adults and healthcare providers’’. ( line122). Is it still ongoing or concluded?

2. The authors need to throw more light on the computer-assisted telephone interviewing data collection instrument bearing in mind that the sub-Saharan African countries involved in the study are developing countries with adolescents having little or no access to computers or phones in rural and urban areas.

3. In addition, adolescent range from 10-19 years. Considering the social economic factors listed in this article, these adolescents may not be able to decipher the different types of vaccines. Parents may also likely be the major determinants of their acceptance or rejection of the vaccines. Kindly clarify how these adolescents were able to decide on their own to accept or reject the vaccine. Adolescents within the ages of 18-19 can be legally considered as adults who can take decisions for themselves. Adolescents between 10-17 years which represent a large percentage of your study are legally not able to decide for themselves to accept or reject covid-19 vaccines. I believe that a strict categorization of these adolescents into smaller age brackets would have provided a more reliable data for this study. Willis et al and Afifi et al as referenced in your article demonstrated these sub-divisions. ( lines 530,535).

4. Furthermore, a sample size of 300 households showed the data was collected from adolescents living in these households. Considering that this study was conducted between July and December, 2021 which falls into the school calendar. Adolescents are expected to be in school because the adolescent age is a school age for children and young adults. Kindly clarify how you reached these adolescents in both urban and rural areas. How was data collected from uneducated adolescents in rural/urban areas?

5. There are several countries in sub-Saharan Africa, the authors may wish to clarify how they determined the samples and sizes more scientifically.

6. In the study, data was collected from those above the defined adolescent age. The study reported in some cases data was collected from adolescents aged 20years as at the time of round 2 data collection (line 160,161). I believe the authors could have maintained the 10-19years adolescent age range.

7. Repeated references was alluded to “the previous round 1 survey” and “ the details of the study design and study population in round 1 described previously(lines 151, 178,205,206)”. Is this study a continuation of another research?

8. To give this study an independent outlook, I believe the authors could have concisely outlined the relevant information from “the round 1” which was relevant to round 2 to avoid confusion due to incessant referrals to read up a previous article.

9. A clearly outlined aims and objectives at the beginning of the study could further complement the discussions and conclusions from this study. .

10. The authors identified three major limitations of this study (lines 442-453). These give the impression that the study areas, sample size determination and data collected did not give a robust representation of the true picture of the covid-19 hesitancy among adolescents in sub-Saharan Africa.

CONCLUSION: I agree with the authors that the strength of this article is the inclusion of different cities from multiple countries in sub-Saharan Africa and the attempt to show a wide array of the potential determinants of Covid-19 vaccine hesitancy among adolescents. In addition, the claim by the authors that this study is the first of its kind to examine the levels and determinants of covid-19 vaccine hesitancy among adolescents in multiple countries from sub-Saharan Africa makes this study unique and novel.

Finally, this article shows a massive work done with good editing and outlay of data with statistical analysis. The study data as presented by the authors clearly support the conclusions drawn by the study.

6. PLOS authors have the option to publish the peer review history of their article (what does this mean?). If published, this will include your full peer review and any attached files.

**Do you want your identity to be public for this peer review?** For information about this choice, including consent withdrawal, please see our Privacy Policy.

Reviewer #1: **Yes: **Bola Christie ATULOMAH (PhD)

Reviewer #2: No

---

## [Editor Report · Decision Letter 1]

30 Aug 2022

COVID-19 vaccine hesitancy and its determinants among sub-Saharan African adolescents

PGPH-D-22-00807R1

Dear Dr. Wang,

We are pleased to inform you that your manuscript 'COVID-19 vaccine hesitancy and its determinants among sub-Saharan African adolescents' has been provisionally accepted for publication in PLOS Global Public Health.

Best regards,

Nnodimele Onuigbo Atulomah, PhD

Academic Editor

Congratulations for having successfully followed the reviewers' suggestions to revise the manuscript. I am therefore recommending this manuscript for possible publication.